# eHealth Literacy in German Skin Cancer Patients

**DOI:** 10.3390/ijerph19148365

**Published:** 2022-07-08

**Authors:** Henner Stege, Sara Schneider, Andrea Forschner, Thomas Eigentler, Dorothée Nashan, Svea Huening, Frank Meiss, Saskia Lehr, Martin Kaatz, Robert Kuchen, Katharina C. Kaehler, Maximilian Haist, Jutta Huebner, Carmen Loquai

**Affiliations:** 1Department of Dermatology, University Medical Center Mainz, 55131 Mainz, Germany; sara@schneider-bombach.de (S.S.); maximilian.haist@unimedizin-mainz.de (M.H.); carmen.loquai@unimedizin-mainz.de (C.L.); 2Department of Dermatology, University Hospital Tuebingen, 72076 Tuebingen, Germany; andrea.forschner@med.uni-tuebingen.de; 3Department of Dermatology, Venerology and Allergology, Charité-Universitätsmedizin Berlin, Corporate Member of Freie Universität Berlin and Humboldt-Universität, 10117 Berlin, Germany; thomas.eigentler@live.com; 4Department of Dermatology, Hospital Dortmund, 44137 Dortmund, Germany; dorothee.nashan@klinikumdo.de (D.N.); svea.huening@klinikumdo.de (S.H.); 5Department of Dermatology, Medical Center—University of Freiburg, 79104 Freiburg, Germany; frank.meiss@uniklinik-freiburg.de (F.M.); saskia.lehr@uniklinik-freiburg.de (S.L.); 6Department of Dermatology, Wald-Klinikum Gera, 07546 Gera, Germany; martin.kaatz@srh.de; 7Institute of Medical Biostatistics, Epidemiology and Informatics (IMBEI), University Medical Center, 55131 Mainz, Germany; robert.kuchen@unimedizin-mainz.de; 8Department of Dermatology, Venerology and Allergology, University Hospital Schleswig Holstein, Campus Kiel, 24105 Kiel, Germany; kkaehler@dermatology.uni-kiel.de; 9Klinik für Innere Medizin II, Hämatologie und Onkologie, Universitätsklinikum Jena, 07747 Jena, Germany; jutta.huebner@med.uni-jena.de

**Keywords:** eHealth, skin cancer, eHealth literacy, health-related information

## Abstract

The global incidence of skin cancer has steadily increased in recent years, and malignant melanoma still has one of the fastest-growing incidence rates among all malignant tumors in the western world. Thus, newly diagnosed patients have an increased need for health information concerning their disease. Using a standardized questionnaire, our study aims to investigate our patients’ primary sources of health-related information as well as their self-proclaimed eHealth literacy. We received 714 questionnaires. Regardless of age, the primary source of information was the treating dermato-oncologist, followed by the treating general practitioner and the Internet. However, with increasing age, the usage of the Internet decreased. Hence, younger participants were better equipped to find health-related information while using the Internet. Additionally, comprehending health-related information and gaining medical knowledge was significantly increased in better-educated participants. Overall, our study shows that with increased use of eHealth services, accessing web-based information increased, correlating with a better eHealth literacy of our patients. eHealth technologies are increasingly becoming more prevalent as a primary source of information in our modern health care system. Thus, it is crucial to educate cancer patients in eHealth literacy to make autonomous, informed decisions and gain more confidence in dealing with their disease.

## 1. Introduction

The incidence of skin cancer has increased dramatically in recent decades, and melanoma still has one of the fastest-growing incidence rates among all malignant tumors in the western world [1,2]. Thus, these patients have an increased need for health information concerning their disease [3,4]. Naturally, patients look for information regarding prognosis, treatment options, associated side effects, and second opinions [5,6]. Still, the treating physician/oncologist remains the primary source of information [7]. However, time restraints occurring in the regular clinical routine often impede a satisfactory response and might not sufficiently meet all requirements of cancer patients [8,9]. Thus, patients turn to other means in order to gather information. While the established sources of information such as books, journals, or support groups remain an essential source of information, the use of the Internet to garner information is constantly increasing since it offers a convenient, around-the-clock, and cost-efficient way to collect and access health information. The access to the more traditional sources of health information can be challenging due to limited financial resources, social isolation, or functional impairment [10,11]. Various studies have shown that cancer patients use the Internet to collect information on multiple subjects [12,13], mainly to interpret symptoms, find second opinions, or seek emotional support [14,15]. Thus, the Internet has become one of the essential tools for cancer patients to seek and find information regarding their disease [16,17]. Especially in the industrialized world, the accessibility of the Internet through the high volume of cell phones and computer use has dramatically increased. Accordingly, over 5 billion people, representing nearly 65% of the global population, are Internet users [18,19,20]. Consequently, the Internet has removed many barriers to access health information [21,22]. However, the broad offers of information can be challenging and difficult to filter and comprehend, especially since there is often a discrepancy between the accessibility and quality of information, which then, in turn, can lead to wrong decisions [23,24].

Thus, patients must develop specific skills to filter, understand and appraise health information. The ability to understand, appraise, and apply health information is summarized in “eHealth literacy” [25], which consists of six different literacy skills: traditional, health, information, scientific, media, and computer literacy. One’s ability can be measured through the eHealth literacy scale [20,26]. In the last decade, the delivery of health-related information has shifted from a more traditional face-to-face exchange into a web-based mode due to the increasing use of digital devices. However, web-based health-related information among older cancer patients is irregular since various barriers such as difficulty navigating the Internet, lack of trust in the provided information, and privacy/security concerns limit its accessibility [27,28]. Furthermore, recent studies have demonstrated that lower eHealth literacy is often associated with a significant risk of hospitalization due to lower usage of preventive services, reduced psychosocial health, and increased episodes of anxiety [29]. Moreover, high eHealth literacy has been associated with increased empowerment and improved survival rates [30]. Nevertheless, data regarding the relationship between self-perceived eHealth literacy and an individual’s ability to use health information are currently missing. Therefore, it is crucial to determine the eHealth literacy of patients in order to provide optimal ways and means for medical information. In this study, we aimed to answer the following research questions:(1)What are the primary sources of health-related information for German skin cancer patients?(2)Are German skin cancer patients satisfied with the provided health-related information?(3)What factors play a role in the self-perceived eHealth literacy of German skin cancer patients?

## 2. Materials and Methods

This was an anonymous survey using a standardized questionnaire. The questionnaire was derived from the first version of a questionnaire on information needs of cancer patients and the Internet [31] and was conceived by experts from the group Prevention and Integrative Oncology of the Working group German Cancer Society [31]. We further modified questions regarding eHealth literacy and eHealth use. Inclusion criteria were diagnosed skin cancer and age >18 years of age. In addition, patients with language restrictions or other limitations that prevented an independent completion of the questionnaire and patients under 18 years of age were excluded from participation.

### 2.1. Research Participants

We surveyed participants who attended their regular follow-ups in 6 German skin cancer centers, including Mainz, Tübingen, Kiel, Gera, Dortmund, and Freiburg. All participants received a standardized questionnaire before attending their follow-up examination and were asked to return the questionnaire at the end of their consultation voluntarily. All participants agreed to participate in this survey voluntarily and signed informed consent.

### 2.2. Research Methods

The questionnaire comprised data on the following subjects:Demographic data (patient during or after treatment, gender, age, type of cancer, year of diagnosis, education level).Data on use of the Internet (focusing on frequency, accessibility and Internet connection). To obtain a better comparability, we categorized the Internet usage of the participants in two groups “heavy user” (“several times a week”/“daily”) and “occasional user” (“several times a month”/“rarely”/“never”) in accordance with Halwas et al., 2017 [32].Data on the overall satisfaction with the currently provided information (cause and progression of the disease, treatment options, possible side effects, comprehensibility of the information) via a Likert scale (1 = dissatisfied to 5 = satisfied). However, the participants did not receive specific eHealth information from their skin center. Accordingly, participants were asked to rate the information available, irrespective of the source of information.Data on the general usage of the Internet, eHealth usage (e.g., electronic devices, apps).Data on eHealth literacy (regarding understanding, finding, secureness, usefulness, reliability, and increased knowledge provided by web-based information). These questions were compiled based on the eHealth Literacy Scale [32].Data on Single Item Literacy Screener.

For this questionnaire, solely closed questions were used. Therefore, a list of possible answers was provided (e.g., “how often did you use the Internet in the last three months?”: daily, several times a week, several times a month, rarer, never). In case of questions requiring a self-rating, the participants answered on a Likert scale (e.g., “How would you rate your understanding of health-related Internet offers?”: 1 = I disagree to 10 = I agree).

### 2.3. Ethics Statement

The multicenter questionnaire was approved by the Ethics Committee of Rheinland-Pfalz (837.385.17) and was conducted in accordance with the principles of the Helsinki Declaration in its current version.

### 2.4. Analysis

The statistical evaluation was carried out with IBM SPSS statistics version 23. GraphPad Prism version 5 was used for data collection. Correlations were determined with the help of the chi^2^ test; *p* < 0.05 was considered significant. In addition, a new significance level was determined using Bonferroni correction when applying several tests to one data set: In the case of categorical and metric variables, we have used the student’s *t*-test (if there were two items as a categorical variable, or ANOVA test if our data showed more than two items as categorical variables) to check for significance. If two different categorical variables were to be compared, we used the Chi-square test. For the rare case that a categorical variable had to be compared with an ordinal scaled variable, we used Kruskal–Wallis, Fisher’s Exact Test, or Mann–Whitney U Test (significance level at *p* < 0.05.).

## 3. Results

### 3.1. Demographic Data

A total of 714 patients in six German skin cancer centers participated in this survey. Among these, 292 (40.9%) were female, and 360 (50.4%) were male, while 62 (8.7%) did not disclose any information on gender. The average age of the participants was 61.81 years (SD 14.54, range 18–89 years). More than three-quarters (79.3%) of the participants were older than 51 years, while only 20.7% were younger than 51 years. Thus, we categorized the participants into three age subgroups: <51 years (20.7%), 51–65 years (32.6%), and >65 years (46.6%) [32]. The education level of the participants was classified into low and high. Hence, participants without a degree, elementary school, general school degree, or secondary school diploma were categorized as participants with low education levels (62.7%). Participants with high school diplomas and university degrees were classified as participants with high education levels (36.1%).

The majority of participants were diagnosed with malignant melanoma (76.9%; *n* = 514), followed by basal cell carcinoma (*n* = 41), cutaneous lymphoma (*n* = 31), and squamous cell carcinoma (*n* = 28) We categorized participants into two subgroups according to melanoma stages: loco-regional (melanoma in situ, stage I and II) and advanced melanoma (stage III and IV) (Table 1).

### 3.2. Source of Information

Regardless of age, the most often reported source of information was the treating dermato-oncologist (*n* = 526), followed by the general practitioner (*n* = 374) and the Internet (*n* = 301) (Table 2). Internet usage as a primary source declined with the increasing age of the participants (Fisher’s Exact Test *p* = 0.052). Conversely, participants over 65 years used their general practitioner as an information source more often than younger participants (Chi-square test *p* = 0.043). Regarding gender or melanoma stage, no significant differences regarding the primary source of information were noticeable (Mann–Whitney-U-Test *p* = 0.139). However, female patients named their practicing dermato-oncologist more often as a primary source of information than male participants (Kruskal–Wallis-Test *p* = 0.032). Books and print media were also sources independent of the participant’s education level (Figure 1).

### 3.3. Level of Satisfaction with the Provided Medical Information

On a Likert scale from 1 (totally dissatisfied) to 5 (very satisfied), the participants were asked to rate their level of satisfaction regarding the provided information, as described in the methodic part of this manuscript (Table 2).

Regarding the provided information on the “Cause of the disease”, the mean value was 3.9. Interestingly, female participants yielded a significantly higher mean value (4.0) in comparison to their male counterparts (3.77) (Mann–Whitney U test *p* = 0.016). Moreover, the provided information was interpreted significantly more favorably with increasing age (Kruskal–Wallis test *p* < 0.001). Additionally, participants with diagnosed melanoma scored higher mean values of 3.87 than those with non-melanoma skin cancer3.61 (Mann–Whitney U test *p* = 0.004).

The level of satisfaction with the information regarding the progression of the disease scored a mean value of 4.0. There was no statistically significant difference in gender, level of education, or cancer entities. However, participants diagnosed with melanoma were slightly more satisfied with the provided information (3.99) than those with non-melanoma skin cancers (3.77), albeit without statistical significance. Notably, patients showed higher overall satisfaction with the provided information with increasing age (Kruskal–Wallis test *p* < 0.001).

The information content of the different therapy options yielded a mean value of 4.24, scoring one of the highest values of all items (see Table 3). There was no significant statistical difference correlating with gender or education. However, participants with diagnosed melanoma had significantly higher mean values of 4.18 than those with non-melanoma skin cancer, 3.83 (Mann–Whitney U test *p* = 0.005). Again, we observed that the overall satisfaction with the provided information increased with age (Kruskal–Wallis test *p* = 0.002).

When assessing the comprehensibility of the provided information, the mean value was 3.91. No significant difference in mean values was found for gender, educational level, or tumor entities. Participants with diagnosed melanoma were significantly more satisfied than participants with non-malignant skin cancers (*p* = 0.045).

Information regarding the treatment medication, including possible adverse events yielded a mean value of 4.6. Participants with melanoma yielded a significantly higher mean value of 4.37 than those with non-melanoma skin cancer entities 4.04 (Mann–Whitney U test *p* = 0.03). Furthermore, we observed an age-related increase in the mean value (Mann–Whitney U test *p* = 0.001).

### 3.4. Usage of the Internet and Other Electronical Devices

A vast majority of the participants (*n* = 505, 73.9%) reported Internet usage within the last three months. In addition, nearly half of the participants (*n* = 337, 49.13%) reported using the Internet daily, while 19.24% (*n* = 132) of the participants never used the Internet (Figure 2).

In total, 67.5% of the participants were classified as “heavy users”, while 32.5% were assigned to the group of “occasional users”. Furthermore, we observed a statistical difference in the frequency of Internet usage regarding gender (*p* = 0.004). Hence, 72.2% of the male participants reported frequent Internet usage compared to 61.4% of the female participants. Concerning age groups, we observed significant difference (*p* = <0.0083); thus, 93.6% of the younger participants (<51 years of age) were classified as “heavy users”, while only 45.3% of the participants over 65 years of age reported a frequent Internet use (Figure 3). We observed significantly higher Internet use in participants with higher levels of education than in participants with lower levels of education (*p* = <0.0001). Nevertheless, we also observed a significant correlation between the age of the participants and their level of education (*p* < 0.0001 Accordingly, we found that 53.6% of the participants under 51 years had a higher level of education, while only 25.6% of participants older than 65 earned a higher level of education. Subsequently, we tested for significance within the different subgroups according to the age of the participants (*p* < 0.001).

Most participants (73.5%) rated their Internet connection as sufficient, while only 4.3% (*n* = 29) reported that their Internet connection was not fast enough for regular use. Interestingly, nearly 15% of patients (*n* = 102) documented that they did not have access to an Internet connection at all. In addition, we observed an age-related decrease in the quality of the Internet connection; additionally, this subgroup had the highest number of participants without an Internet connection. Additionally, the availability of a sufficient Internet connection significantly increased in correlation to the participant’s education level; we reason that the social-economic status of the participants plays a significant role in the availability of a sufficient Internet connection. (*p* = <0.001). In general, most participants accessed the Internet via a computer/laptop either from their home (74%) or their workspace (26%). Cell phones or other mobile devices were used by 62.9% of the participants.

### 3.5. eHealth Usage

Altogether, 413 participants out of 488 answered the question about the usage of electronic health services and offers. However, most participants (*n* = 175, 35.8%) reported an infrequent usage, while only 7.9% (*n* = 39) confirmed a daily use. A third of the participants (31.5%, *n* = 154) would not consider using eHealth services, 19.6% (*n* = 95) of the participants reported using eHealth offers once, and 21.3% (*n* = 104) did not yet use electronic health services but would hypothetically do so.

We observed a significant difference regarding gender in the usage of eHealth offers (*p* = 0.0008). Specifically, 45% of the female participants frequently used eHealth offers compared to 33.7% of the male participants. We did not find a significant difference in the usage of eHealth offers regarding age (*p* = 0.932) or education (*p* = 0.09). However, eHealth offers were more often used in the subgroup of participants with a lower level of education than in the subgroup with a high level of education (41.4% vs. 35.4%), albeit without statistical significance. To address eHealth habits, participants were asked about their utilization of different eHealth services. For example, 80.3% of participants have accessed the Internet to search for health information, followed by 49.5% seeking medical terms in an online encyclopedia, and 14.3% searching for medical specialists. Modern eHealth tools such as Apps, fitness trackers, or eBooks were rarely used in our patient cohort (see Figure 4).

We could not detect significant differences between female and male participants with regard to the frequency of accessing the Internet to gather health-related information. However, patients with higher education levels (*p* = 0.004) and advanced melanoma (*p* = 0.003) used self-help groups significantly more often. Further, online encyclopedias, journals, and brochures were significantly more often used by participants with a higher level of education.

### 3.6. Data on eHealth Literacy

To assess the eHealth literacy of the participants, the following items were gathered by a Likert scale from 1 (I disagree) to 10 (I agree) as described in the methodic part of this manuscript (see above).

First, the participants were asked to rate their level of “Comprehension of health-related information provided by the Internet”, and 16.5% of the participants were confident in fully comprehending Internet information. On average, the participants rated their comprehension level as 6.94. Regarding gender and age, we did not observe a significant difference in the comprehension levels of the participants. In addition, the ability to comprehend health-related information was statistically significantly lower in patients with higher levels of education in contrast to patients with a lower level of education (*p* = 0.001). Concerning the clinical melanoma stage, no significant differences between the subgroup of participants with loco-regional or advanced melanoma were detectable (6.4 vs. 6.6 *p* = 0.242) (Figure 5).

Next, the participants evaluated their ability to find relevant information regarding their health and disease. The arithmetic mean value was 6.57. Again, we did not find a significant difference regarding gender or age. However, there was a significant difference in the subgroup of participants with higher levels of education (7, 4) in comparison to those with a lower level of education (6, 34) (*p* = 0.0034).

Furthermore, finding relevant information significantly increased in participants who reported frequent Internet usage or other electronic devices.

The next item examined the participant’s sense of certainty when making a health-related decision based on information from the Internet. Compared to the other arithmetic mean values, this item yielded the lowest overall score of 3.77. No statistical association was observed concerning gender, education levels, or melanoma stage no statistical association was observed. We observed that patients of higher age showed a significantly higher sense of certainty when making a decision based on information provided by the Internet (*p* = 0.008) (Figure 6).

Due to the overflow and constant availability of health-related information, patients’ main difficulty is discerning which information is reliable or appropriate. Thus, the next item examined whether the participants were confident in allocating health-related information to reliable and unreliable sources. We recorded here the second lowest mean value 3.92. Notably, 19.2% had no certainty when allocating sources into reliable and unreliable information (participants who previously stated no Internet use were excluded) We did not notice a statistically significant difference concerning gender, level of education, melanoma stage, or age.

The item aimed to assess if health-related information via the Internet increased the medical knowledge of our participants yielded a mean value of 5.92. While there was no statistical association concerning gender, we observed a statistical association regarding age (*p* = 0.016) and level of education (*p* = 0.001). In terms of age, we found that older participants and those with higher education levels rated their increase of medical knowledge more favorably than younger participants.

In the last two items, we examined the usage of mobile health applications in our participant cohort. Here, we assessed if the participants could allocate suitable apps for their daily health care needs. Patients were moderately confident finding apps that could help their disease (mean value: 5.17). Noteworthy, 13.4% had no confidence in their ability to allocate such apps. Interestingly, the older subgroups rated their ability to find suitable apps significantly lower in comparison to the younger participants (<51 years: 5.55, 51–65 years: 5.24, >65 years: 4.71; *p* = 0.041). Additionally, the association between the level of education and the ability to find suitable apps was statistically significant. Thus, participants with higher education levels yielded a mean value of 5.63, while those with low education levels reached a 4.86 (*p* = 0.006).

Secondly, we assessed the practicability of using those apps through smartphones or tablets. The arithmetic mean value was 5.36; however, 12.6% of the participants reported that they could not apply eHealth apps via smartphones or tablets appropriately. While there was no significant association regarding gender, we observed that in our younger subgroups, the ability to use eHealth apps was significantly increased (*p* = 0.001). In addition, participants with higher education levels had higher confidence in their ability to use eHealth apps.

### 3.7. Single Item Literacy Screener (SILS)

The SILS is an effective tool to identify the patients with limiting reading and comprehension ability and therefore require assistance reading health-related materials. Possible answers were: 1—never, 2—rarely, 3—sometimes, 4—often, and 5—always [33]. Most of our participants stated that they never (35.2%) or rarely (31.2%) needed assistance when reading health-related materials, while 21.4% sometimes required help. We classified participants with SILS scores >2 as patients with “limited reading ability” [32]; thus, 30.1% of the participants fell into this category. No significant associations regarding gender or melanoma stage were observed. Not surprisingly, there was a significant association between the SILS, education (*p* = 0.001), and age (*p* = 0.006). Accordingly, 34.7% of the participants with lower levels of education had SILS scores higher than 2 and were therefore classified as patients with “limited reading ability”. In comparison, only 21.4% of the participants with higher levels of education were grouped into this category. Additionally, we observed an amplification of participants with SILS scores >2 with increasing age (<51 years: 18.8%, 51–65 years: 33.8% and >65 years: 32.8%). Interestingly, frequent eHealth services and eHealth literacy usage was not significantly associated with lower SILS scores.

## 4. Discussion

The incidence of skin cancer has increased dramatically in recent decades. Hence, an increasing number of patients have to be informed about their disease and possible treatment options. Recent literature suggests that the Internet is developing into one of the primary sources of health-related information. In particular, it has previously been demonstrated that the Internet is becoming the primary source for health-related information in younger patients on the one hand. On the other hand, for patients of older age, established sources in our health care system (e.g., general practitioners, treating dermato-oncologist) remain the constant and favored vehicle to obtain health-related information.

Furthermore, recent clinical observations suggest an association between eHealth literacy and Internet use. Therefore, it has been proposed that patients with low eHealth are often not skilled enough in searching online for health-related information as well as to differentiate between reliable and unreliable. To our knowledge, this is the first prospective study that used an internally consistent eHealth literacy scale to determine the use of the Internet and other online eHealth services for German skin cancer patients. Here, we obtained several vital findings that corroborate previous concepts of the role of eHealth literacy in acquiring and understanding health-related information from the Internet.

First, our results revealed that physician–patient interaction remains the primary source of information among our participants. However, the use of the Internet as a primary source of information for health issues rapidly increases in younger patients [14,34,35]. We observed an age-dependent decline in the use of the Internet [32,34]. Moreover, with increasing age, significantly more male participants used the Internet. Similar observations were made by Zickhur et al.; in American adults. Here, nearly half of the adults 65 and older remain disconnected from the Internet. Except for computers, seniors aged 65 and older are less likely than other age groups to own any digital devices [36], while the typical Internet-using student uses the Internet for 100 min per day and over 99% possess a digital device [37].

With increasing age, the Internet was used less frequently, and consultation through patient–physician interaction increased accordingly. Our data is consistent with prior published data that show an increase in patient–physician exchange as the primary source in patients over the age of 75 and with lower educational backgrounds. Primary reasons could be attributed to the unfamiliarity of daily Internet use, which complicates searching, filtering, and gathering online information and frequently requires a certain level of external help to recognize relevant information [5,32,38]. Accordingly, in the last two decades, the use of the Internet increased from 29% in 2000 to nearly 94% in 2020 [39]. In comparison, 80.8% of our study participants used the Internet at least occasionally. We attribute the discrepancy in Internet use to the difference in the average age of our participants (61.55 years) as compared to the younger age of the overall German population (44.5 years) (Bundesinstitut für Bevölkerungsforschung (BiB), 2021).

Second, we demonstrated that older participants appeared to be significantly more satisfied with the currently provided information about their disease in terms of cause, progression, or treatment options than their younger counterparts. The increased satisfaction regarding the provided information in older participation might be explained through a preexisting relationship with the treating physician. Thus, newly obtained information might be viewed as trustworthy. However, it must be noted that our survey did not evaluate the treating physicians or the patient–physician interaction. In contrast, younger patients were less satisfied with the provided information from various outlets, most often the Internet. Thus, one could deduce a higher need for information at a younger age, skepticism towards the information received, and a need for “better information”. Again, our data is consistent with previous studies [40,41], which found that the Internet became the primary source of information, especially in younger melanoma patients. At the same time, patients were much more satisfied when receiving most of the information on their disease from the treating physician than getting similar information from the Internet. Our findings indicate that the Internet is gaining relevance as a primary source of information; however, it has not replaced the patient–physician interaction due to the lack of reliability and, for some patients, a lack of accessibility. Hence, the treating physician remains the most trusted and most widely used source of information for German skin cancer patients [42]. However, the increasing use of online information could benefit the patient and promote shared decision-making by strengthening patient autonomy and improving the physician–patient relationship [5].

Additionally, we could demonstrate that the use of eHealth services is significantly higher in female participants, while age or education did not influence the use of eHealth services. Our findings are in accordance with previous literature that indicates that women are more likely than men to play an active role in obtaining information and to seek information from as many different sources as possible [43,44].

Furthermore, we observed the lowest mean value when evaluating if eHealth services assisted patients when making health-related decisions, followed by the ability to differentiate between reliable and unreliable sources. However, allocating the information seems to be not the primary problem since participants rated their ability to find relevant information with a mean value of 6.57. Therefore, it seems that patients are skeptical about online health-related information [45]. Moreover, due to the multitude and complexity of the information, they do not have enough security to make decisions about their health based on this information. One explanation might be that younger patients are more aware of unreliable sources [5,39,46]. Although the additional certainty provided by the information from the Internet may help patients deal with their disease, this observation also carries the risk of wrong decisions being made based on unreliable and non-evidence-based sources.

Additionally, we observed that participants with lower education were significantly more often uncertain while using health-related information and making decisions based on this information. Participants with lower education rated their ability to find and differentiate eHealth-related information significantly lower than their counterparts. Additionally, they showed a poorer understanding and use of eHealth services. Accordingly, participants with higher education used eHealth offers regularly and reported that the application expanded their medical knowledge. In summary, participants with a lower level of education are struggling to deal with electronic health services, presumably due to a lack of competence in handling these services. Thus, these patients show a lower eHealth literacy. Hence, it is vital to reinforce efforts to increase eHealth literacy in these subgroups, as any patient may benefit from information on their disease. Our findings align with current publications by Schaeffer et al. that low educational status correlates with low health literacy. Thus, these deprived subpopulations are severely challenged by the increased requirements for accessing, understanding, and applying health-related information from the Internet. Additionally, in recent years there has been an increase in misinformation and interest-driven information [47].

Last, our results revealed that nearly one-third of the participants were categorized into the category “limited reading ability” and that these patients showed the lowest levels of competence in the handling of electronic health services. This observation complements a recent study, which found similar results [32], suggesting that participants with adequate reading ability benefit from eHealth-related information. Unfortunately, further investigation of the limited reading ability was not possible, so possible confounders such as limited vision, illiteracy, or poor language skills cannot be ruled out.

When interpreting the results of our study, additional limitations to consider are the various Internet sources the patients may have used to gather disease-related information. It would therefore be important to include the source of Internet information gathered by the patients in order to allow for a more comprehensive and differentiated analysis of Internet-related health information. Additionally, as of the multicenter aspect of this survey the exact information provided from the treating dermato-oncologists might have differed between the skin cancer centers which might have affected the patents’ answers in the survey.

Furthermore, following our previous results, participants with low education levels were more likely to have limited literacy concerning medical instructions, brochures, or other written material. Thus, literacy, eHealth literacy, and education level all seem to impact each other. It should also be noted that the literature not only describes a dependency between eHealth literacy and educational level [48,49] but also on prognosis and health status [50,51]. Therefore, integrating eHealth literacy into general education should be critical since the Internet will likely develop into the primary source for health-related information. However, it will be necessary to improve the accessibility and handling skills in the daily use of the Internet via infrastructure concepts and community-based education programs, particularly for older people, who were found to have a lower level of competence in dealing with eHealth services and the Internet.

## 5. Conclusions

We provide evidence that the Internet has become one of the most sought-after tools for obtaining information about younger patients. Thus, it is of the utmost importance to increase eHealth literacy so that the information can be interpreted correctly. Thus, enabling an equal partnership in the physician–patient interaction and genuinely establishing a shared decision making. In particular, we could show that younger and better-educated participants could use health-related information in their decision-making process by allocating information to reliable or unreliable sources. Furthermore, one-third of the participants had inadequate reading abilities, highlighting the association between eHealth literacy and education. Hence, educating cancer patients in eHealth literacy is crucial to increasing their ability to make autonomous, informed decisions, thus gaining more confidence in dealing with their disease. Based on our observations, further research is needed to identify the needs of our patients in terms of health-related information and thus improve their eHealth literacy skills.

## Figures and Tables

**Figure 1 ijerph-19-08365-f001:**
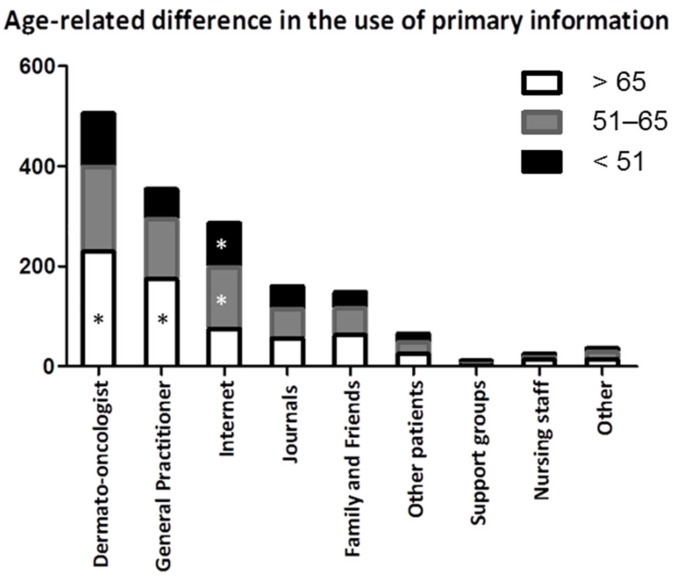
Age-related difference in the use of primary information. Older participants received information significantly more often from their dermato-oncologist and general practitioner. Abbreviations: * *p* < 0.05.

**Figure 2 ijerph-19-08365-f002:**
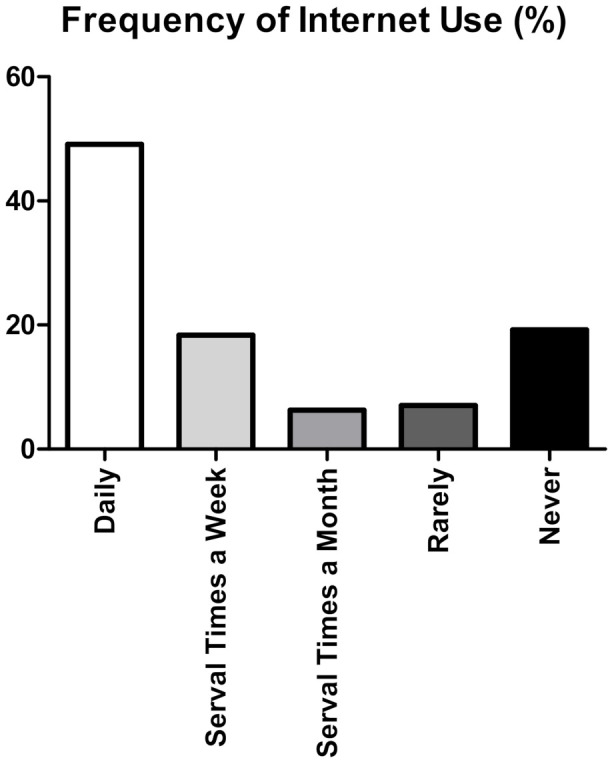
Frequency of Internet use. Descriptive bar chart showing the frequency of Internet use in our cohort. Most of the participants (<65%) reported frequent Internet usage. Nonetheless, 19.24% (*n* = 132) of the participants never use the Internet.

**Figure 3 ijerph-19-08365-f003:**
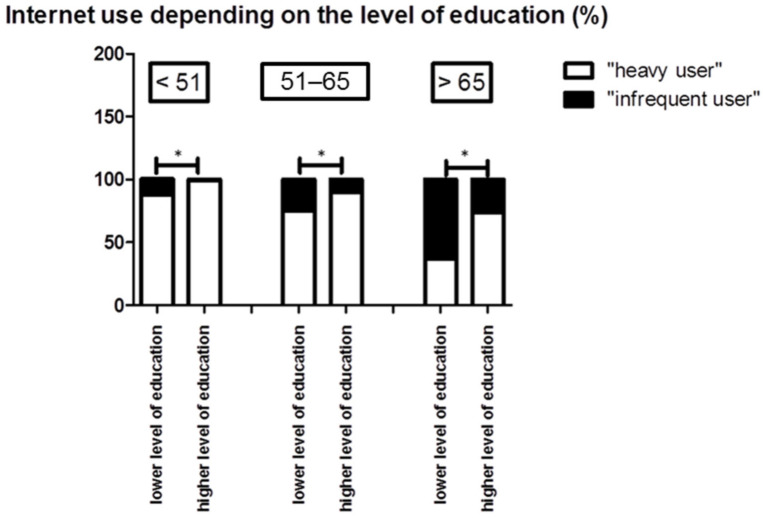
Internet use depending on the level of education. Younger patients and patients with a higher level of education more frequently used the Internet (*p* = 0.05). Abbreviations: * *p* < 0.05.

**Figure 4 ijerph-19-08365-f004:**
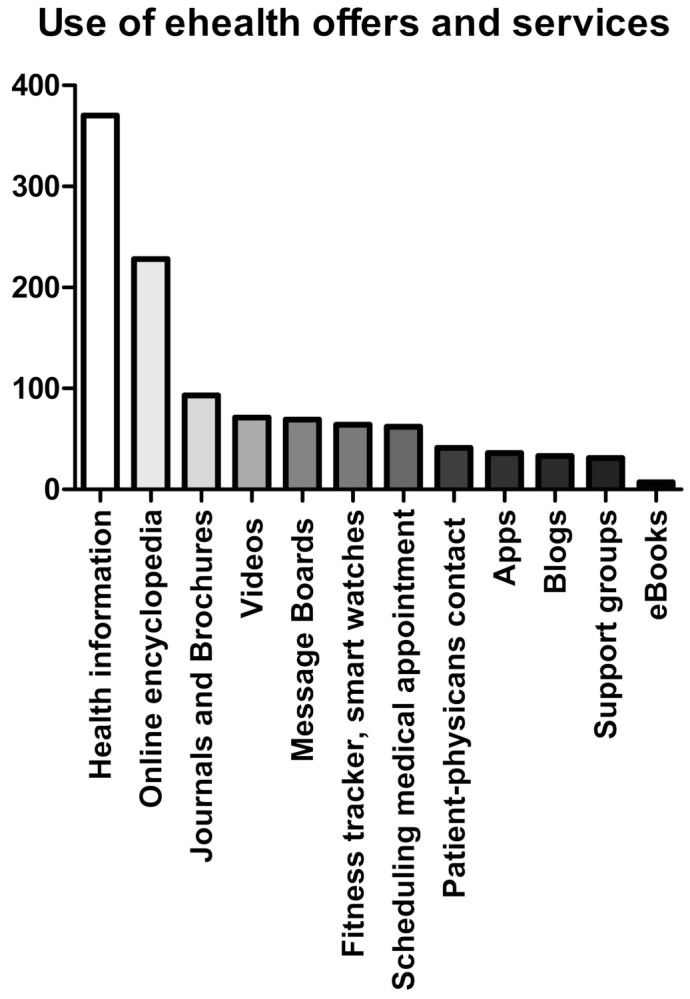
Use of eHealth offers and services. Total number of participants using eHealth offers and services categorized by the different outlets.

**Figure 5 ijerph-19-08365-f005:**
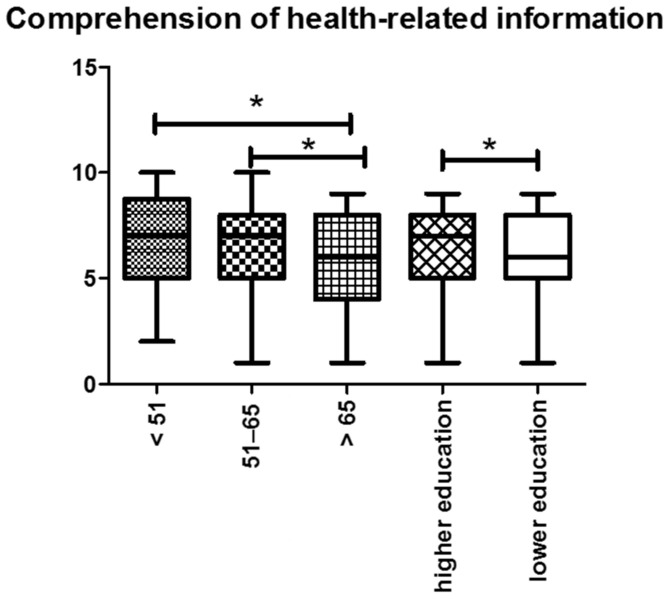
Comprehension of health-related information: We noticed a decreasing arithmetic mean value with increasing age of the participants. Additionally, the ability to comprehend health-related information was statistically significant in patients with higher levels of education. Abbreviations: * *p* < 0.05.

**Figure 6 ijerph-19-08365-f006:**
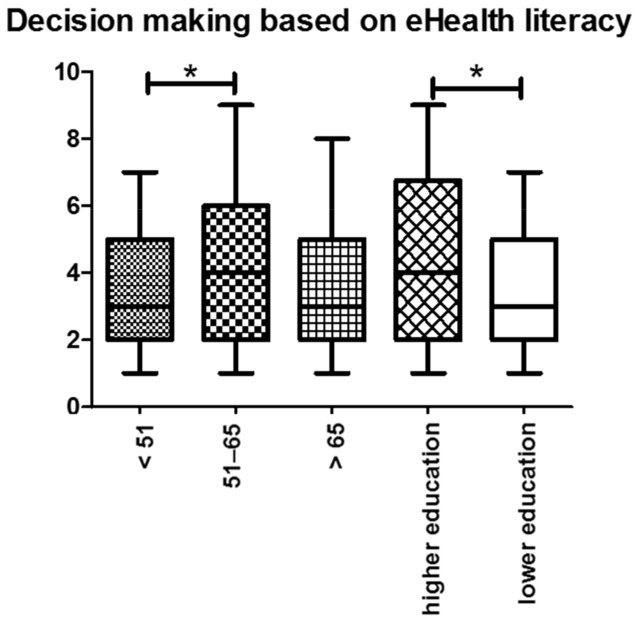
Decision making based on eHealth literacy: In general, older participants were more certain when making a decision based on health-related information from the Internet. Similarly, participants with higher education levels were more certain in their decision-making process. Abbreviations: * *p* < 0.05.

**Table 1 ijerph-19-08365-t001:** Baseline characteristics of participants.

	*n* (%)
Age (years)	61.81 (range 18–89)
<50	26 (4%)
51–65	116 (17%)
>65	324 (48%)
Gender	
Female	292 (40.9%)
Male	360 (50.4%)
No data	62 (8.7%)
Tumor entities	514 (77%)
Malignant melanoma	41 (6.1%)
Basal cell carcinoma	31 (4.6%)
Cutaneous lymphoma	28 (4.1%)
Squamous cell carcinoma	22 (3.2%)
Mycosis fungoides	15 (2.2%)
Merkel cell carcinoma	10 (1.4%)
Cutaneous sarcoma	3 (0.4%)
Melanoma stages (AJCC 2017)	
Melanoma in situ	3 (0.6%)
I	129 (25.1%)
II	66 (12.8)
III	142 (27.6)
IV	148 (28.8)

**Table 2 ijerph-19-08365-t002:** Primary source of information.

	*n* (%)
Dermato-oncologist	526 (31.1%)
General practitioner	374 (22.1%)
Internet	301 (17.8%)
Journals	167 (9.9%)
Family and friends	159 (9.4%)
Other patients	68 (4.0%)
Nursing staff	27 (1.6%)
Support groups	11 (0.7%)
Others	57 (3.4%)

(*p* = 0.047), while younger participants significantly more often received their information via the Internet (*p* = 0.03). No significant differences were observed for other sources of information.

**Table 3 ijerph-19-08365-t003:** Level of satisfaction with the provided information.

	Mean Score	Women	Men	*p*-Value	<51	51–65	>65	*p*-Value	Melanoma	Non-Melanoma Skin Cancers	*p*-Value	Stage I, II	Stage III, IV	*p*-Value
Cause of the disease	3.9	4.0	3.77	**0.016**	3.56	3.84	4.10	**<0.001**	3.87	3.61	**0.004**	4.01	3.87	0.18
Progression of the disease	4.0	4.06	3.91	0.105	3.64	3.94	4.18	**<0.001**	3.99	3.77	**<0.001**	4.03	3.97	0.55
Treatment options	4.24	4.30	4.14	0.108	4.04	4.18	4.35	**0.035**	4.18	3.83	**0.002**	4.39	4.17	0.039
Comprehensibility of the information	3.91	4.05	3.94	0.896	3.94	4.06	4.18	**0.022**	3.95	3.89	0.64	4.31	4.07	**0.023**
Medication	4.6	4.67	4.55	0.31	4.60	4.67	4.59	0.77	4.37	4.04	**<0.001**	5.16	4.12	**<0.001**

Statistics based on the Likert scale score of the different items regarding the satisfaction with the provided information. The *p*-value is indicated in bold numbers when statistically significant.

## Data Availability

Data will be made available upon request.

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
