# Peer review of "eHealth Literacy in German Skin Cancer Patients"

_ijerph, 2022, doi:10.3390/ijerph19148365_

Round 1

Reviewer 1 Report

IJERPH  eHealth literacy in German skin cancer patients

An interesting and important subject is studied in this article (with many authors), addressing possible eHealth inequality resulting from eHealth literacy.

 Some typo’s I highlighted in the pdf version attached.

Major remarks:

General:

The Methods section and Results are not fully aligned, nor is there a specific research question on which the Methods can be ‘checked’ on adequacy. The Results show more specific figures than the Methods suggests. Most importantly though, the research questions should be more clear to start with.

1) Introduction: Goal is said to be:

“To learn more about the health literacy of melanoma patients and their usage of the Internet, we conducted a survey in six different German skin 71 cancer centers using a questionnaire”.

Is the paper indeed about Health literacy (as there are no questions in the survey on Health literacy), or about eHealth literacy?

Although the importance of the subject is clear from the introduction, I miss a specific research question and hypotheses. In the Results part several comparisons are made, whereas there are no questions or hypotheses formulated on which these results give an answer on.

2) Methods:

The introduction suggests only melanoma patients are involved, but table 1 shows otherwise. Note: Better to give table 1 in the Results section

As the exact purpose of the study is not clear, the method cannot be valued in an optimal manner. In the results, there are many comparisons and associations, while there is no introduction on the aim of these.

For instance: which are the dependent and independent variables? (Also related to the research question(s))

There were several hospitals involved: is their information provision comparable, as the data are not analysed separately? I can imagine that this might influence the answers in the survey. When results are given on comprehensibility of information, the source also is important. (Now it is mentioned ‘information from the internet’,  which might also be specifically and reliably offered by (some of the) hospitals, or other health care providers.

I would expect (after reading the Results): Questions on demographics (gender, SES, social network, age and other factors associated with digital technology use), Health Literacy (see aim), eHealth literacy, ad use related Qs, but why also satisfaction? (Is there a hypothesis on this?) Overall (again), the paper would benefit from a clear aim, hypotheses and consistency in introduction & methods, with clear dependent and independent (outcome) variables. Reasoning back from the results, different research Qs could be formulated, with a clear position on age, as the Results are described for different age categories.

Can the authors provide the full survey as well to better value the Method? (supplementary material)

As many of the variables used are interdependent, why do the authors not use multiple regression analysis: what is the relative contribution of, e.g., education, age and gender on the different outcome variables?

3) Results

The texts give statistical values, but the tables (e.g. table 2) do not provide this information.

3.5: in the Methods it is not described that eHealth usage is included in the survey.

What is meant with eHealth offers and how is the practical situation: do the different hospitals provide eHealth opportunities?

The results present ‘data on eHealth literacy’,  which comprise ‘certainty on decisions based on Internet information’.  I cannot retrieve this in the methods. E.g., ‘Thus, the next item examined whether the participants were confident in their ability to allocate 290 health-related information into reliable and unreliable sources.’ As the items are not described, it is not clear what is meant here, nor how to interpret this item.

In the results SILS is mentioned, which should be (also) part of the Methods.

4) Discussion

“We reasoned that the increased level of satisfaction can be attributed to the fact that older participants more often obtain their information from health care specialists”à I think this cannot be concluded, as this was not part of the results.

The same conts for: ‘Since participants rated their ability to find relevant information with a mean value of 6.57. Our data rather indicate that the lack of high-quality information might complicate an informed decision-making process for patients, since health-related information on the Internet frequently lacks crucial basic information, thus providing incorrect or incomplete knowledge…’

This conclusion is speculative, as this relationship was not part of the study or not presented as such).

Author Response

1: Introduction: Goal is said to be:

To learn more about the health literacy of melanoma patients and their usage of the Internet, we conducted a survey in six different German cancer centers using a questionnaire”..

Is the paper indeed about Health literacy (as there are no questions in the survey

on Health literacy), or about eHealth literacy?

Although the importance of the subject is clear from the introduction, I miss a

specific research question and hypotheses. In the Results part several comparisons

are made, whereas there are no questions or hypotheses formulated on which these results give an answer on.

Ad 1: We agree with the reviewer that a clear research question was initially not postulated. Accordingly, we have changed the introduction of our manuscript and formulated research questions regarding the primary source for health-related information and if German skin cancer patients are satisfied with the current state of the provided information. Last we tried to determine what factors play a role in the self-perceived eHealth literacy of German skin cancer patients.

2: The introduction suggests only melanoma patients are involved, but table 1 shows otherwise. Note: Better to give table 1 in the Results section

Ad 2: We agree with the reviewer that initially, the wording of our introduction suggested that only melanoma patients participated in this survey. Accordingly, we changed the introduction to skin cancer as the overarching tumor entity. Thus, we hope that the interested reader understands that we included patients with various skin cancers.

3: As the exact purpose of the study is not clear, the method cannot be valued in an optimal manner. In the results, there are many comparisons and associations, while there is no introduction on the aim of these.

For instance: which are the dependent and independent variables? (Also related to the research question(s))

Ad 3: We thank the reviewer for this valuable hint and agree with the notion that our section on Materials and Method has to be revised in order to clarify our observations. We have therefore provided a more detailed analysis of the inclusion and exclusion criteria, the setting in which the participants took part in this survey. Furthermore, we have added a more in depth description of our questionnaire and a detailed description of our statistical procedure.

  1. There were several hospitals involved: is their information provision comparable,

as the data are not analysed separately? I can imagine that this might influence the answers in the survey. When results are given on comprehensibility of information, the source also is important. (Now it is mentioned ‘information from the internet’, which might also be specifically and reliably offered by (some of the) hospitals, or other health care providers.

Ad. 4: We share the concern of the reviewer that due to the multicentre aspect of this survey, the provided information may differ from one skin center to the other. Unfortunately, it is impossible to determine precisely if our participants received similar information from their treating dermato-oncologist. Our questionnaire aims to get a general overview of the satisfaction with the information sources already available. Likewise, only closed questions were asked, so we cannot go into further detail on specific sources.

5: I would expect (after reading the Results): Questions on demographics (gender, SES, social network, age and other factors associated with digital technology use), Health Literacy (see aim), eHealth literacy, ad use related Qs, but why also satisfaction? (Is there a hypothesis on this?) Overall (again), the paper would benefit from a clear aim, hypotheses and consistency in introduction & methods, with clear dependent and independent (outcome) variables. Reasoning back from the results, different research Qs could be formulated, with a clear position on age, as the Results are described for different age categories.

Ad 5: Again, we are in agreement with the reviewer, that our section on Material and Methods needed a revision in order to clarify the aim of our research and to proper introduce the different items of our questionnaire. We hope that by revising the section on material and method, our research objectives will be clearer and easier for the reader to assess.

6: Can the authors provide the full survey as well to better value the Method? (supplementary material)

As many of the variables used are interdependent, why do the authors not use multiple regression analysis: what is the relative contribution of, e.g., education, age and gender on the different outcome variables?

Ad 6: We will be happy to provide the questionnaire if required. Unfortunately, we cannot publish the raw data due to data protection laws, as it is possible to identify a partially unidentified person.

7: The texts give statistical values, but the tables (e.g. table 2) do not provide this information.

Ad 7: We have edited the manuscript so that the statistical values are consistent in the text and the figures/graphs.

 8: 3.5: in the Methods it is not described that eHealth usage is included in the survey.

Ad 8: We have modified the section on Material and Methods accordingly, so that the item on eHealth usage is now included.

9: What is meant with eHealth offers and how is the practical situation: do the different hospitals provide eHealth opportunities?

Ad 9: This survey aims not to examine the different information services offered by skin cancer centers. The participating skin cancer centers did not evaluate their information services separately. Various online offers relating to skin cancer treatment are described as eHealth offers.

10: The results present ‘data on eHealth literacy’, which comprise ‘certainty on decisions based on Internet information’. I cannot retrieve this in the methods. E.g., ‘Thus, the next item examined whether the participants were confident in their ability to allocate 290 health-related information into reliable and unreliable sources.’ As the items are not described, it is not clear what is meant here, nor how to interpret this item.

Ad 10: In order to better evaluate the results on self-assessed eHealth literacy and SILS, the corresponding items of the questionnaire were explicitly presented again in our material and method section. We hope that this will lead to a better interpretation of the results.

11: “We reasoned that the increased level of satisfaction can be attributed to the fact that older participants more often obtain their information from health care specialists”à I think this cannot be concluded, as this was not part of the results.

Ad 11: Our survey showed that older participants received their information directly from their dermato-oncologist, while younger participants used information from the internet more often.

In addition, older patients showed higher general satisfaction with the current information. Accordingly, we assume that the communication channel via the treating physician leads to increased trust and thus higher satisfaction.

12: The same conts for: ‘Since participants rated their ability to find relevant information with a mean value of 6.57. Our data rather indicate that the lack of high-quality information might complicate an informed decision-making process for patients, since health-related information on the Internet frequently lacks crucial basic information, thus providing incorrect or incomplete knowledge…’

Ad 12: We agree with the reviewer that our statements were hypothetical and could not be confirmed by our results. Accordingly, we have removed this statement and focused on distinguishing between reliable and unreliable sources as this issue is the biggest challenge for patients' eHealth literacy.

Reviewer 2 Report

The article titled ‘eHealth literacy in German skin cancer’ may be an useful contribution to the journal; however, few changes should be taken into consideration:

Introduction is sound and offers reasonable context and the rationale for the following reseach presented in the manuscript.

Lines: 25-26: We received 714 questionnaires – authors should state out of how many, as this information is relevant. A workflow needs to be presented (questionnaires intended, questionaries responded, how many patients did not respond)

‘after treatment’ patients are somehow more prone to finding and researching for information and to absorbe the information provided by the physician, taking the skin condition more seriuously as the number of medical visits progresses.

Hence, it is expected that ‘after’ patients are more informed than ‘1st visit’ patients. This should be reflected somehow (or at least analysed) in the results to check for differences in groups.

Lines 49-50: While the established source of information such as books, journals, or support groups  remain an essential source for information, the use of the Internet for information is increasing rapidly  - authors should make a short estimation or to ellaborate on the reason that fundament this situations (such as, but not limited to, for example, prohibitive prices for  non-open-access articles or for books.)

Matherial and Methods: authors do not explicitly state how the patients were selected: were they consecutive patients or maybe some other criteria were involved? This is important, in order to minimise selection bias.

Ethics approval and ethics committee should be named explicitly.

Figures 5 and 6 should include either box-whickers plots or at least to include standard deviations bars, not to mention that an asterisc * is needed to be inserted where pairwise comparisons do show statistically significant values; in current form, it it absolutely impossible to assess if the difference between values in columns is statistically significant, just by eye-balling the charts, therefore the reader cannot understand if the values throughout the several groups (e.g. age groups) differ; also tables with median/mean, st dev, p values should be included in the manuscript, in order to present to the interested reader, clearly and in an academic manner, in a more eloquent way, the results of the study.

Grammar and punctuation must also be carefully checked within the entire manuscript.

Author Response

  1. We received 714 questionnaires – authors should state out of how many, as this Information is relevant. A workflow needs to be presented (questionnaires intended, questionaries responded, how many patients did not respond).

Ad 1: We agree with the reviewer that it is desirable to accurately count the number of questionnaires distributed, followed by an enumeration of those completed and those not completed. Unfortunately, due to the multicentre design of this survey and the fact that it was carried out during regular outpatient consultations, such documentation was not possible.

  1. ‘after treatment’ patients are somehow more prone to finding and researching for information and to absorbe the information provided by the physician, taking the skin condition more seriuously as the number of medical visits progresses.

Hence, it is expected that ‘after’ patients are more informed than ‘1st visit’ patients. This should be reflected somehow (or at least analysed) in the results to check for differences in groups.

Ad 2: We agree with the reviewer that patients on therapy certainly understand their disease better and, thus, are likely more skilled in searching for information regarding their diagnosis. Unfortunately, our questionnaire does not allow us to draw any conclusions about the patients' treatment state. However, all patients have already been diagnosed with cutaneous neoplasia and connected to a German skin center. Therefore, it can be assumed that almost all patients are not presenting for the first time and that the cohort is mainly homogeneous.

  1. Lines 49-50: While the established source of information such as books, journals, or support groups remain an essential source for information, the use of the Internet for information is increasing rapidly - authors should make a short estimation or to ellaborate on the reason that fundament this situations (such as, but not limited to, for example, prohibitive prices for non-open-access articles or for books.)

Ad 3: We agree with the reviewer that our statement that the internet is gaining weight as a primary source of information should be further discussed. Accordingly, we have discussed the advantages the internet has compared with more traditional sources (availability, open access).

  1. Matherial and Methods: authors do not explicitly state how the patients were selected: were they consecutive patients or maybe some other criteria were involved? This is important, in order to minimise selection bias.

Ad 4: We thank the reviewer for this valuable hint and agree with the notion that our section on Materials and Method has to be revised in order to clarify our observations. We have therefore provided a more detailed analysis of the inclusion and exclusion criteria, the setting in which the participants took part in this survey. Furthermore, we have added a more in depth description of our questionnaire and a detailed description of our statistical procedure.

  1. Ethics approval and ethics committee should be named explicitly.

Ad 5: We have added to ethics approval to the manuscript.

  1. Figures 5 and 6 should include either box-whickers plots or at least to include standard deviations bars, not to mention that an asterisc * is needed to be inserted where pairwise comparisons do show statistically significant values; in current form, it it absolutely impossible to assess if the difference between values in columns is statistically significant, just by eye-balling the charts, therefore the reader cannot understand if the values throughout the several groups (e.g. age groups) differ; also tables with median/mean, st dev, p values should be included in the manuscript, in order to present to the interested reader, clearly and in an academic manner, in a more eloquent way, the results of the study.

Ad 6: We agree with the reviewer's concerns that due to an inadequate statistical presentation, the interested reader cannot fully assess the results of our study. Thus,  we have revised the corresponding graphs and added box-whickers plots. In addition, we have revised our tables and added appropriate nomenclature (e.g., mean value, p values). We hope that this will enable a better interpretation of our results by the interested reader.

7.Grammar and punctuation must also be carefully checked within the entire manuscript.

Ad 7: The manuscript has been reviewed and corrected by a native speaker in order to enable a more comprehensible reading experience. We hope we have fulfilled the required changes in our manuscript and incorporated the suggested revisions to your full satisfaction in order to fully meet your requirements for publication.   

Round 2

Reviewer 1 Report

The manuscript has been improved. Three aspects can benefit from change:

- Ad remark 4: Because of the importance of different information provided, this needs to be addressed in the Discussion as a possible weakness or flaw. 

- Ad remark 6: Please do add the survey for the readers. Note: not the raw data are asked, the Q was about the statistical analysis. This Q has not been answered yet.

- Ad remark 9: Explanation about the eHealth offered should be given.

- Ad remark 11: As the results do not show (and were not intended to show) the satisfaction about their physician, this conclusion should be drawn with less certainty. The explanation can be other, e.g.: perhaps older adults tend to be more satisfied on all kinds of subjects. 

Author Response

Thank you very much for the extensive and very helpful comments on our manuscript.

- Ad remark 4: Because of the importance of different information provided, this needs to be addressed in the Discussion as a possible weakness or flaw. 

Ad 4: We thank the reviewer for this important criticism. We have included an additional paragraph discussing the limitations of our survey (L457-471).

- Ad remark 6: Please do add the survey for the readers. Note: not the raw data are asked, the Q was about the statistical analysis. This Q has not been answered yet.

Ad 6: Again, we would like to thank the reviewer for this important remark. Certainly, a multiple regression analysis might help to better determine the relative contribution of variables such as gender, age and education on the outcome variables (i.e., eHealth literacy or overall satisfaction). As for the multitude of other variables that have not been included in the survey (such as general condition, social environment and support, duration of the disease or response to treatments etc) we refrained from multiple regression analysis and rather decided to limit our analysis to the description of our observations on the impact of single variables on the investigated outcomes.  

 - Ad remark 9: Explanation about the eHealth offered should be given.

Ad 9: The participants did not receive any individual information about their disease from the treating skin center. Accordingly, the question of our questionnaire is aimed at the general information available to the respective participant. Therefore, we have added a corresponding passage in the section for material and methods, in order to clarify the goal of our questionnaire and avoid inconsistencies in the interpretation of our results.

- Ad remark 11: As the results do not show (and were not intended to show) the satisfaction about their physician, this conclusion should be drawn with less certainty. The explanation can be other, e.g.: perhaps older adults tend to be more satisfied on all kinds of subjects. 

Ad 11: We agree with the reviewer that increased satisfaction with the existing information service is not exclusively due to the patient-doctor relationship, as this was not elicited in our questionnaire. Therefore, we have revised our discussion accordingly and pointed out that evaluating the patient-doctor relationship was not part of the survey. Hence, another possible aspect to consider is the increase of chronic illnesses in older patients. Thus, these patients are more often confronted with information regarding an unknown disease and, therefore, might have an increased level of comprehensibility in evaluating health information.

Reviewer 2 Report

The manuscript has been improved I have no further comments.

Author Response

Thank you very much.

This manuscript is a resubmission of an earlier submission. The following is a list of the peer review reports and author responses from that submission.